# Impact of Early Pancreatic Graft Loss on Outcome after Simultaneous Pancreas–Kidney Transplantation (SPKT)—A Landmark Analysis

**DOI:** 10.3390/jcm10153237

**Published:** 2021-07-22

**Authors:** Lukas Johannes Lehner, Robert Öllinger, Brigitta Globke, Marcel G. Naik, Klemens Budde, Johann Pratschke, Kai-Uwe Eckardt, Andreas Kahl, Kun Zhang, Fabian Halleck

**Affiliations:** 1Department of Nephrology and Medical Intensive Care, Charité-Universitätsmedizin Berlin, 10117 Berlin, Germany; marcel.naik@charite.de (M.G.N.); klemens.budde@charite.de (K.B.); kai-uwe.eckardt@charite.de (K.-U.E.); andreas.kahl@charite.de (A.K.); fabian.halleck@charite.de (F.H.); 2Department of Surgery, Charité-Universitätsmedizin Berlin, 13353 Berlin, Germany; robert.oellinger@charite.de (R.Ö.); brigitta.globke@charite.de (B.G.); johann.pratschke@charite.de (J.P.); 3Berlin Institute of Health (BIH), 10117 Berlin, Germany; kun.zhang@charite.de; 4Department of Internal Medicine and Cardiology, Campus Virchow-Klinikum, Charité-Universitätsmedizin Berlin, 13353 Berlin, Germany

**Keywords:** simultaneous kidney–pancreas transplantation, kidney transplantation, pancreas transplantation, diabetes mellitus type I, graft loss

## Abstract

(1) Background: Simultaneous pancreas–kidney transplantation (SPKT) is a standard therapeutic option for patients with diabetes mellitus type I and kidney failure. Early pancreas allograft failure is a complication potentially associated with worse outcomes. (2) Methods: We performed a landmark analysis to assess the impact of early pancreas graft loss within 3 months on mortality and kidney graft survival over 10 years. This retrospective single-center study included 114 adult patients who underwent an SPKT between 2005 and 2018. (3) Results: Pancreas graft survival rate was 85.1% at 3 months. The main causes of early pancreas graft loss were thrombosis (6.1%), necrosis (2.6%), and pancreatitis (2.6%). Early pancreas graft loss was not associated with reduced patient survival (*p* = 0.168) or major adverse cerebral or cardiovascular events over 10 years (*p* = 0.741) compared to patients with functioning pancreas, after 3 months. Moreover, kidney graft function (*p* = 0.494) and survival (*p* = 0.461) were not significantly influenced by early pancreas graft loss. (4) Conclusion: In this study, using the landmark analysis technique, early pancreas graft loss within 3 months did not significantly impact patient or kidney graft survival over 10 years.

## 1. Introduction

Simultaneous pancreas–kidney transplantation (SPKT) is considered the best therapeutic option for patients with diabetes mellitus type I and kidney failure [1,2]. When performed successfully, the patient will be free from dialysis and independent from insulin, with a significant improvement in quality of life, lower rates of comorbidities, and better survival, as has been shown in many studies [3,4,5,6,7,8]. However, risks resulting from the surgical procedure and the lifelong immunosuppression have to be taken into account and weighed against the potential benefits. With advancing technologies and refined immunosuppressive regimens, SPKT has become safer, with fewer complications [9,10,11,12]. However, early pancreas graft failure is still rather frequent, with an occurrence of more than 10% according to registry data [1,13,14]. The literature provides inconsistent data for its impact on various outcomes. While several studies in the past showed that early pancreas graft failure is associated with increased kidney allograft loss and worse outcomes [15,16,17,18], a very recent study reported the contrary [19]. In addition, previous study results may be influenced by time-dependent effects, since pancreas allograft failure occurs over time. A landmark analysis would be the preferred statistical technique to address this concern [20].

The aim of this study was to perform a landmark analysis to assess the impact of early pancreas graft loss within 3 months on kidney graft function and patient survival over 10 years after SPKT. 

## 2. Materials and Methods

### 2.1. Patient Selection and Data Acquisition

In this retrospective single-center study, we included all 114 adult patients who consecutively received an SPKT between January 2005 and December 2018 at the transplant center Charité Campus Virchow Klinikum in Berlin. All patients had diabetes mellitus type I and had to fulfill the Eurotransplant eligibility criteria, as Germany is part of Eurotransplant. These only include patients with negative C-peptide levels (<0.5 ng/mL) or diabetes type I-specific antibody positivity. SPKT in diabetes mellitus type II patients is only allowed in strictly exceptional cases in the Eurotransplant region [21]. Our electronic patient database “TBase” [22], clinical charts, and the Eurotransplant Network Information System (ENIS) were used for the data collection. Key outcomes (patient and graft survival, renal function) were observed over a period of up to 10 years post-transplant, yielding a complete data set without any patients lost to follow-up. The study was approved by the institutional Ethics Committee (EA4/202/19).

### 2.2. Immunosuppression

Primary immunosuppression in all SPKT recipients (SPKTR) was based on a triple-drug regimen with tacrolimus (Astellas, Tokyo, Japan), mycophenolate (MPA), either mycophenolate mofetil (MMF) (Roche, Basel, Switzerland) or enteric coated mycophenolic sodium (ECMPS) (Novartis Pharma, Rotkreuz, Switzerland), and steroids. Tacrolimus was initially dosed at 0.15 mg/kg/d, and trough levels were maintained at 8–10 ng/mL for 6 months and at 5–7 ng/mL afterwards. Initial daily MMF dosage was 2 g/day. All SPKTRs received induction therapy with a lymphocyte-depleting agent (antithymocyte globulin (ATG)) (Sanofi-Aventis, Paris, France). In case of contraindications for ATG, an IL-2R antagonist (basiliximab) (Novartis Pharma, Rotkreuz, Switzerland) was used.

### 2.3. Surgical Technique 

Donor pancreas and kidney were procured from deceased donors with adequate organ quality for combined kidney pancreas transplantation. All organs were procured as donations after brain death (DBD), since organs from donation after circulatory death (DCD) are not available in Germany. The pancreas was placed in the abdomen using caval drainage for venous outflow. Arterial inflow from the right common iliac artery was established using a Y graft of the donor’s iliac bifurcation to the superior mesenteric artery (SMA) and splenic artery, and enteric drainage of exocrine secretions was established using a duodenojejunostomy. The kidney was placed in the recipient’s left iliac fossa with vascular anastomoses to the iliac vessels and ureterocystoneostomy.

### 2.4. Anticoagulation Procedures

In the immediate post-transplant period, patients were treated with unfractionated heparin (UFH) with a targeted activated partial thromboplastin time (aPTT) of 45–50 s to prevent vascular thrombosis of the graft. In cases of partial thrombosis detected on routine ultrasound examinations or by computer tomography (CT), which is usually performed on clinical indication, patients were treated with elevated targeted aPTT of 60–80 s or with therapeutic weight-adjusted doses of low-molecular-weight heparin to prevent further apposition.

### 2.5. Definitions

Kidney graft failure was defined as return to maintenance dialysis, allograft nephrectomy, or re-transplantation [23]. Delayed kidney graft function (DGF) was defined as receiving at least one dialysis procedure (hemodialysis or peritoneal dialysis) within the first week post-transplant. Borderline findings and acute rejections (BANFF ≥ Ia or ABMR) on biopsy were recorded according to the BANFF 2013 criteria [24]. For calculation of kidney donor profile indices (KDPI) for deceased donors, the Organ Procurement and Transplantation Network (OPTN) calculation and mapping table referring to the median donor of 2017 were used [25,26]. The pancreas donor risk index (PDRI), indicating pancreas donor quality, was calculated by the formula published by Axelrod et al. in 2010 [27]. Pancreatic rejection findings on biopsy were recorded and graded according to the updated BANFF grading scheme [28]. Pancreatic graft failure was defined as return to insulin dependency. The cause of pancreas graft loss was reported based on pathologic explantation reports. Major Adverse Cardiac and Cerebrovascular Events (MACCE) were defined as cardiovascular death, non-fatal myocardial infarction, coronary revascularization procedure, or non-fatal stroke. Estimated glomerular filtration rate (eGFR) was calculated by the Chronic Kidney Disease Epidemiology Collaboration (CKD-EPI) formula [29]. For group comparison, a GFR of 0 mL/min was imputed to account for kidney graft failure.

### 2.6. Data Analysis

Donor and recipient characteristics are shown as means with standard deviation or, in case of non-normal distribution, as median with interquartile range (IQR of 25th and 75th percentile). Statistical significance of the differences between groups for continuous variables were tested using student’s t-test or Mann–Whitney-U test in case of a non-parametric distribution and chi-squared test for categorical variables. Fisher’s exact test was used to analyze binary data. The significance level was set to α = 0.05. Kaplan–Meier and logistic analyses were performed to determine graft and patient survival with a confidence interval of 95%. The Kaplan–Meier landmark for analyses for kidney graft and patient survival was set at month 3 to avoid time-dependent effects (immortal time bias) [20]. Patient deaths and kidney graft losses within the landmark are listed descriptively. No missing values were substituted, due to a complete data set. IBM SPSS 26 (IBM Germany GmbH, Ehningen, Germany) for Mac was used for statistical analysis.

## 3. Results

### 3.1. Characteristics of Recipients and Donors

Out of 114 patients, 17 patients (14.9%) suffered from early pancreas graft loss within 3 months. Median follow-up time of the whole cohort was 6.6 (IQR 4.1–10.7) years. Recipients of both groups (functioning graft vs. early graft loss) were similar in age (44.4 ± 8.9 vs. 44.7 ± 9.4 years, *p* = 0.931) and BMI (23.4 kg/m^2^ (IQR 21.4–26) vs. 23.4 kg/m^2^ (IQR 20.6–26), *p* = 0.706). Donor age was young for both groups (32.5 ± 10.5 years vs. 31.4 ± 10.2 years, *p* = 0.681). Kidney donor profile indices (KDPI) as well as pancreas donor risk indices (PDRI) were similarly low in both groups (Table 1). In summary, baseline characteristics and immunosuppression of both groups did not differ significantly (Table 1 and Table 2), except for frequency of peritoneal dialysis.

### 3.2. Pancreas Graft Survival

We evaluated pancreas graft loss in the overall cohort during a follow-up period of 10 years with complete follow-up (Figure 1). In this period, 23 patients experienced pancreas graft loss (20.2%). Out of the 23 patients, 14/23 (60.9%) cases occurred within 1 month, 17/23 (73.9%) within 3 months, and 18/23 (78.2%) within 1 year post-transplant. Pancreas graft survival rates censored for death were 85.1% (CI95% 78.6–91.6) at 3 months, 84.2% (CI95% 77.5–90.9) at 1 year, 80.9 (CI95% 73.5–88.3) at 5 years, and 78.0% (CI95% 69.8–86.2) at 10 years. Pancreas rejection episodes in the first year post-transplant were significantly more frequent in the early graft loss group than in the functioning pancreatic graft group (5/17 (29.4%) vs. 5/97 (5.2%) patients, *p* = 0.001). The main causes of pancreas graft loss were thrombosis (6.1%), necrosis (2.6%) and pancreatitis (2.6%). Other causes included chronic pancreatic graft failure, chronic rejection, bleeding, mycotic aneurysm (Figure 2).

### 3.3. Impact of Early Pancreas Graft Loss on Patient Survival

The Kaplan-Meier curve for patient survival over 10 years with the landmark set at month 3 showed no significant difference between the patients with functioning pancreas grafts (78.1%; CI95% 67.9–88.3) and the patients with early pancreas graft loss within 3 months (56.7%; CI95% 26.5–86.9; *p* = 0.168) (Figure 3). Patient survival rates were 98.9% (CI95% 96.7–100) in the functioning pancreas group vs. 87.5% (CI95% 71.2–100) in the early graft loss group at 1 year and 88.8% (CI95% 81.9–95.7) vs. 87.5% (CI95% 71.2–100) at 5 years post-transplant, respectively. Within the landmark period, 3 and 1 deaths occurred in the functioning pancreas graft group and early graft loss group, respectively.

### 3.4. Impact of Early Pancreas Graft Loss on Kidney Graft Survival

Kidney graft survival over 10 years with the landmark set at 3 months was similar in both groups (functioning pancreas: 95.4% (CI95% 89.9–100) vs. early pancreas graft loss: 92.9% (CI95% 79.4–100; *p* = 0.461) (Figure 4). During the time to landmark, two patients in the early pancreas graft loss group and one patient in the functioning pancreas group suffered from primary non-function of the renal allograft.

The incidence of kidney graft failure was overall low. The individual causes are listed in Table 3.

### 3.5. Impact of Early Pancreas Graft Loss on Other Outcomes

Delayed kidney graft function and biopsy-proven rejections occurred at similar rates in both groups. Estimated GFR at 1 year and MACCE at 10 year post-transplant did not significantly differ between patients with pancreas graft loss and functioning graft. As expected, HbA1c was significantly higher in the group of patients with early pancreas loss (Table 4).

## 4. Discussion

This contemporary retrospective single-center study shows that early post-transplant pancreas graft loss (within 3 months) does not significantly affect patient survival or kidney graft function over a follow-up period of 10 years. The purpose of using the landmark analysis technique was to estimate the survival probabilities with no time-to-event bias [20,30]. Based on previous studies [16,17,31] and our own clinical observations that the first 3 months post-transplant represent a vulnerable period of pancreatic graft loss, the landmark was set at 3 months a priori. In fact, technical failures are the most common cause of pancreas graft loss and predominantly occur in the first 3 months after SPKT [16,17,31]. Pancreas survival rates in this study were 85% at 3 months and 84% at 1 year, which is in accordance with previously published results [32,33,34].

In contrast to older studies [16,17], we did not observe that early pancreas graft loss was associated with higher mortality or increased kidney graft loss, which supports the recently published findings by Das et al. [19]. Their data demonstrated good outcomes (in relation to patient survival, hospital length of stay, surgical wound complications, rejection episodes) and excellent kidney allograft function following early pancreas failure. Similar to our study, Das et al. used a contemporary patient cohort with current standard of care. Considering that diagnostic modalities, surgical procedure, perioperative management, and therapeutic options have remarkably improved in the past 30 years, it is plausible that a contemporary patient cohort has better results compared to older studies. 

Thrombosis was the main cause for early pancreas graft loss. Other causes were necrosis, pancreatitis, chronic pancreatic graft failure, chronic rejection, bleeding, and mycotic aneurysm. These results are in line with previous reports [14,16,35]. Despite improvements in the surgical procedure, technical issues remain the most common reason for graft failure. Together with technical issues, coagulation management remains key for success [36]. Although in our cohort early graft loss patients showed higher rates of pancreas rejection on biopsy than patients in the functioning graft group (29% vs. 5%), only one graft loss could be directly attributed to acute rejection. This suggests that treatment of rejection episodes has become more successful, and acute rejection is a rather rare cause of graft loss in SPKT recipients [37].

Kidney graft failure can occur secondary to complications resulting from loss of the pancreas graft, most commonly during sepsis or systemic inflammatory response syndrome [16]. While infection is a risk factor for graft loss, more recent studies showed that mortality associated with sepsis or septic shock after SPKT and other solid organ transplantations is surprisingly low [38,39,40]. Progress generally achieved in all medical care-related aspects and, specifically, regarding the immunosuppressive regimen as well as sepsis management probably contribute to the ongoing improvements noticed in kidney graft function and survival after SPKT [30]. In our cohort, other risk factors for graft loss such as acute rejection during the first year or delayed graft function were in the usual ranges in both groups. As seen in this study, kidney function at 1 year and kidney graft survival 10 year post-transplant were not significantly affected by whether early pancreas loss occurred, although HbA1c levels were significantly higher in the early pancreas graft loss group. Kidney graft loss was overall very low, corresponding to less than 5% after 10 years. Of note, kidney grafts were of high quality, as indicated by the median KDPI of 23 in our cohort and seemed to be relatively resistant to an adverse metabolic state in the early pancreas graft loss group. Moreover, higher levels of HbA1c were not associated with increased MACCE over 10 years, a result which could potentially change in a larger cohort and/or with an even longer observation period [41].

Importantly, survival for SPKT recipients is superior to that of waitlisted patients who remain on dialysis [42]. However, it has been noticed in a US study that the number of SPKT has declined, despite good outcomes [43]. The reasons are most likely multifactorial, involving changes in recipient demographics, local referral patterns, and other factors such as increasing numbers of living kidney transplantation combined with pancreas after kidney transplantation. In addition, overall organ shortage further limits SPKT [12]. The results of our current study confirm the good long-term outcomes and further encourage the performance of SPKT where applicable. There are even data supporting the expansion of this therapy to appropriately selected patients with diabetes mellitus type II [1,44] and older patients [45,46].

Our study has strengths and limitations. Its strengths include a contemporary cohort with a complete and extensive data set of donors and recipients undergoing a rather uniform treatment regimen. The baseline characteristics of the compared groups were homogeneous. To the best of our knowledge, it is the first time that the impact of early pancreas graft failure on long-term outcomes was analyzed by the landmark analysis technique to avoid immortal-time bias. The study is limited by its retrospective, single-center nature and the number of patients. Due to the small number of patients with early pancreas graft loss, a multivariate analysis on risk factors was not feasible. The choice of the time point to set the landmark can be discussed, as a potential limitation of landmark analyses in general. The composition of the cohort was predominantly Caucasian, and therefore, the results may not be generalizable to other ethnic groups. Moreover, only transplantation after brain death was evaluated, since donation after circulatory death is prohibited in Germany.

## 5. Conclusions

In conclusion, this study using the landmark analysis technique showed that early pancreas graft loss within 3 months does not significantly impact mortality or kidney graft survival and function over 10 years. These findings should encourage the listing of patients with diabetes type I and kidney failure for SPKT.

## Figures and Tables

**Figure 1 jcm-10-03237-f001:**
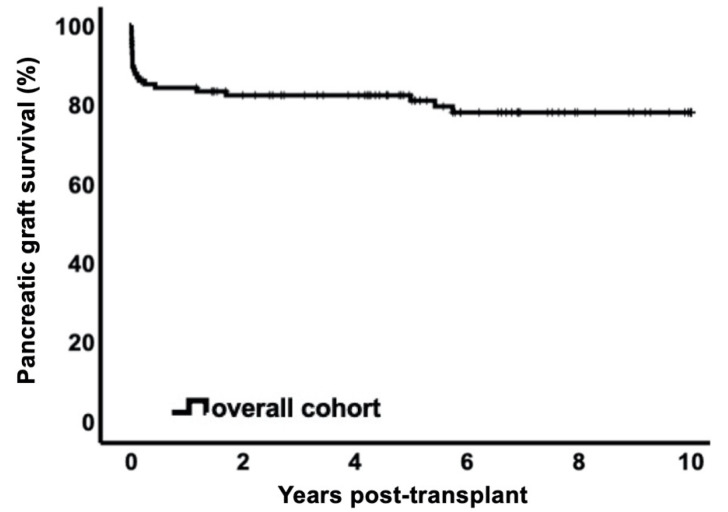
Kaplan–Meier curve for pancreatic graft survival censored for death over 10 years (78.0% (CI95% 69.8–86.2)). Of note, pancreatic graft loss mostly occurred in the first 3 months.

**Figure 2 jcm-10-03237-f002:**
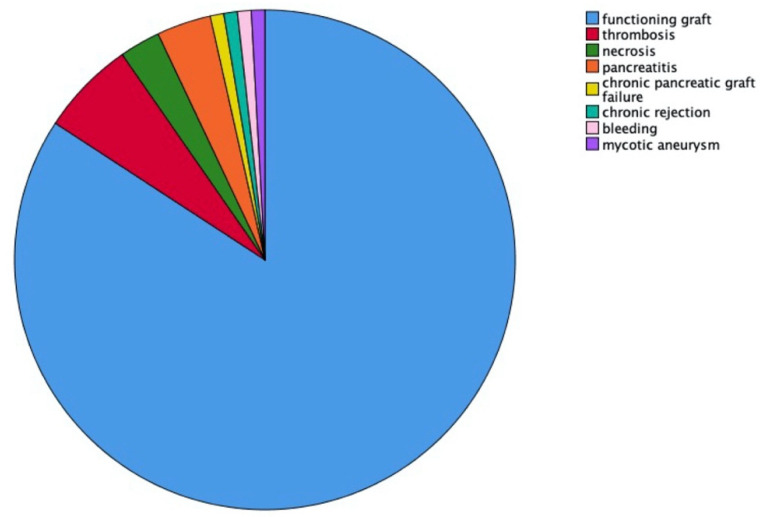
Causes of early pancreas graft loss within 3 months.

**Figure 3 jcm-10-03237-f003:**
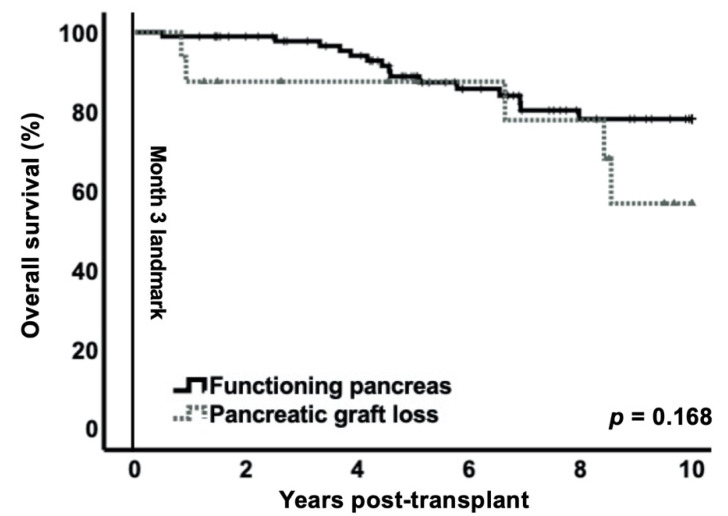
Patient survival over 10 years with 3-month landmark for pancreas graft loss. The Kaplan–Meier curve showed no significant difference between the two groups (*p* = 0.168).

**Figure 4 jcm-10-03237-f004:**
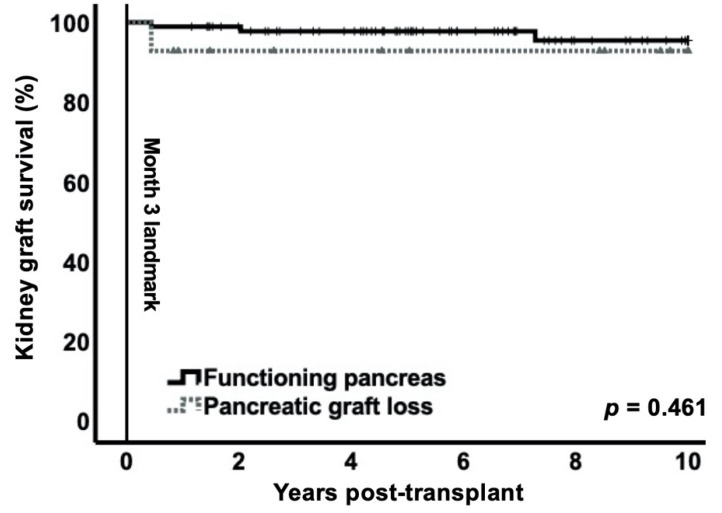
Kidney graft loss over 10 years censored for death with 3-month landmark for early pancreas graft loss. Kaplan–Meier curve showed no difference between the two groups (*p* = 0.461).

**Table 1 jcm-10-03237-t001:** Recipient and donor characteristics.

Variables	Functioning Pancreatic Graft	Early Pancreatic Graft Loss within 3 Months	*p*
**Patients, *n***	97 (85.1%)	17 (14.9%)	
Recipient characteristics			
Mean recipient age, years (SD)	44.4 (8.9)	44.7 (9.4)	0.931
Male, *n* (%)	58 (59.8%)	12 (70.6%)	0.435
Kidney retransplantation, *n* (%)	6 (6.2%)	0 (0%)	0.589
Pancreas retransplantation, *n* (%)	3 (3.1%)	0 (0%)	0.462
Median time on dialysis, months (IQR)	38 (26–53)	28 (19–46)	0.188
Dialysis use (hemodialysis), *n* (%)	67 (69.1%)	16 (94.1%)	0.032
Dialysis use (peritoneal dialysis), *n* (%)	21 (21.6%)	0 (0%)	0.034
Preemptive transplantation, *n* (%)	9 (9.3%)	1 (5.9%)	0.651
Mean HLA mismatches (SD)	4.5 (1.3)	4.4 (0.9)	0.875
Median Cold ischemia time (kidney), hours (IQR)	11 (9.5–12.5)	11 (9.2–12.3)	0.589
Median Cold ischemia time (pancreas), hours (IQR)	9 (7–10.7)	8 (7–10.7)	0.336
Body mass index (BMI), kg/m^2^ (IQR)	23.4 (21.4–26)	23.4 (20.6–26)	0.706
Diabetes Type I (%)	97 (100%)	17 (100%)	-
Mean time since 1st diagnosis of diabetes, years (SD)	29.1 (10.3)	33.0 (9.4)	0.151
Coronary artery disease, *n* (%)	24 (24.7%)	4 (23.5%)	0.916
Arterial hypertension, *n* (%)	96 (99%)	16 (94.1%)	0.163
Prior Stroke, *n* (%)	13 (13.4%)	2 (11.8%)	0.855
Prior myocardial infarction (%)	6 (6.2%)	2 (11.8%)	0.411
HbA1c at transplantation, (%) (IQR)	7.1 (6.1–8)	7 (6.3–7.85)	0.964
Donor characteristics			
Mean donor age, years (SD)	32.5 (10.5)	31.4 (10.2)	0.681
Donor BMI, kg/m^2^ (IQR)	23.1 (21.7–24.7)	23.4 (21.6–25.7)	0.741
Recipient–Donor BMI match index (IQR)	1 (0.91–1.18)	1.02 (0.88–1.14)	0.58
Donor creatinine at transplantation, (mg/dL) (IQR)	0.70 (0.59–0.86)	0.71 (0.64–0.84)	0.386
Kidney Donor Risk Index (KDRI) 2017 (IQR)	0.94 (0.82–1.10)	0.87 (0.79–1.15)	0.706
Kidney Donor Profile Index (KDPI) 2017 (IQR)	23 (10–40)	16 (7–45)	0.717
Mean Pancreas Donor Risk Index (SD)	1.03 (0.26)	1.02 (0.2)	0.841

HLA = Human Leucocyte Antigen; SD = standard deviation; IQR = interquartile range of 25th and 75th percentile.

**Table 2 jcm-10-03237-t002:** Primary immunosuppression after transplantation.

Immunosuppressive Medication	Functioning Pancreatic Graft	Early Pancreatic Graft Loss within 3 Months	*p*
Tacrolimus	97 (100%)	17 (100%)	-
Mycophenolate	97 (100%)	17 (100%)	-
Steroids	97 (100%)	17 (100%)	-
Anti-thymoglobulin (ATG)	92 (94.8%)	16 (94.1%)	0.524
Basiliximab	5 (5.2%)	1 (5.9%)	0.784

- = statistics not applicable.

**Table 3 jcm-10-03237-t003:** Causes of kidney graft failure.

Variables	Functioning Pancreatic Graft *n*	Early Pancreas Graft Loss within 3 Months *n*
Graft thrombosis	1	1
Acute rejection (BANFF IIb)	0	1
Chronic transplant glomerulopathy	1	0
Pyelonephritis	1	0
Infected hematoma	0	1
Septic shock	1	0
Unknown	1	0
Primary non-function	1	2

**Table 4 jcm-10-03237-t004:** Clinical endpoints with functioning pancreas graft vs. early pancreas graft loss.

Variables	Functioning Pancreatic Graft	Early Pancreas Graft Loss within 3 Months	*p*
Patients, *n*	97 (85.1%)	17 (14.9%)	
Delayed graft function	19 (19.6%)	5 (29.4%)	0.364
Kidney biopsy-proven acute rejection (≥BANFF Ia)	12 (12.4%)	2 (11.8%)	0.945
Mean estimated glomerular filtration rate at 1 year post-transplant by CKD-EPI formula censored for death and kidney graft failure (SD)	68.8 (19.2)	64.9 (15.7)	0.494
Imputed eGFR (GFR = 0) at 1 year post-transplant for patients with kidney graft loss by CKD-EPI formula censored for death (SD)	68.1 (20.4)	59.9 (23.4)	0.184
Median HbA1c at 1 year post-transplant censored for death	5.4 (5.1–5.6)	8.1 (6.9–8.6)	<0.001
Major adverse cardiac or cerebrovascular event (MACCE) at 10 years post-transplant, *n*	8 (8.2%)	1 (5.9%)	0.741

SD = standard deviation; CKD-EPI = Chronic Kidney Disease Epidemiology Collaboration; eGFR = estimated glomerular filtration rate.

## Data Availability

The data presented in this study are available on request from the corresponding author. The data are not publicly available due to ethical restrictions.

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
