# Peer review of "Impact of Early Pancreatic Graft Loss on Outcome after Simultaneous Pancreas–Kidney Transplantation (SPKT)—A Landmark Analysis"

_jcm, 2021, doi:10.3390/jcm10153237_

Round 1

Reviewer 1 Report

COMMENTS:

Lehner and Coll. addressed if early post-transplant pancreas graft loss within 3 months affects patient survival or kidney graft function over a follow-up period of 10 years since it is well-known that the risk of pancreas graft loss is highest in the first 3 months post-pancreas transplant. Their retrospective single center study clearly demonstrated that early pancreas graft loss within 3 months did not significantly impact mortality or kidney graft survival and function over 10 years. Data presented in this study are interesting and suggestive of the potentially beneficial effects in pancreas transplantation field

Specific comment

  • Are there any anti-coagulation treatment protocols post pancreas transplant in your program? Please include them in the method sections.
  • Some programs reported that Pancreas US was helpful to detect partial thrombosis, which is treatable. Any protocols?
  • Authors reported that necrosis (2.6%) and pancreatitis (2.6%) were independent causes of pancreas loss. Thrombosis also caused necrosis and pancreatitis. How did you distinguish them?
  • Do authors use C-peptide level for Recipient exclusion criteria? Which level is cut off in your program? Even patients with Type I DM, some of them have positive C-peptide. Do you transplant pancreas to those patients in your program?
  • Has your program done SPK for selected Type II DM patients?

If yes, any differences?

If no, please comment any thoughts regarding SPK for Type II DM patients.

Reviewer 2 Report

This article is a great effort to identify the outcome of patients undergoing SPK. The review is elaborate and well described.
